# Primary Ovarian Leiomyosarcoma Is a Very Rare Entity: A Narrative Review of the Literature

**DOI:** 10.3390/cancers15112953

**Published:** 2023-05-28

**Authors:** Vincenzo Dario Mandato, Federica Torricelli, Valentina Mastrofilippo, Andrea Palicelli, Luigi Costagliola, Lorenzo Aguzzoli

**Affiliations:** 1Unit of Obstetrics and Oncological Gynecology, Azienda USL-IRCCS di, 42122 Reggio Emilia, Italy; 2Laboratory of Translational Research, Azienda USL-IRCCS di, 42122 Reggio Emilia, Italy; 3Unit of Pathology, Azienda USL-IRCCS di, 42122 Reggio Emilia, Italy; 4Unit of Obstetrics and Gynecology, Santa Maria delle Grazie Hospital, 80100 Naples, Italy

**Keywords:** primary ovarian leiomyosarcoma, symptoms, stage, mitotic count, lymphadenectomy, chemotherapy, survival, treatment, review

## Abstract

**Simple Summary:**

Primary ovarian leiomyosarcoma (POLMS) is a very rare malignancy characterized by unclear management and poor survival. We reviewed all 113 cases of POLMS reported in the literature till September 2022 to identify prognostic factors and the best treatment. Most patients received surgical resection, associated with lymphadenectomy in 12.5% of cases. Only 40% of patients received chemotherapy. POLMS is usually diagnosed at an early stage. Increasing stage and number of mitoses are associated with a worse prognosis. On the contrary, surgical resection with lymphadenectomy and chemotherapy is associated with increased survival. Ultimately, 43.4% of patients relapsed, and their mean disease-free survival was 12.5 months. There is a need for an international registry for POLMS that can help collect comprehensive and reliable data from around the world so that the best treatment can be definitively identified.

**Abstract:**

Background: Primary ovarian leiomyosarcoma is a very rare malignancy characterized by unclear management and poor survival. We reviewed all the cases of primary ovarian leiomyosarcoma to identify prognostic factors and the best treatment. Methods: We collected and analyzed the articles published in the English literature regarding primary ovarian leiomyosarcoma from January 1951 to September 2022, using PubMed research. Clinical and pathological characteristics, different treatments and outcomes were analyzed. Results: 113 cases of primary ovarian leiomyosarcoma were included. Most patients received surgical resection, associated with lymphadenectomy in 12.5% of cases. About 40% of patients received chemotherapy. Follow-up information was available for 100/113 (88.5%) patients. Stage and mitotic count were confirmed to affect survival, and lymphadenectomy and chemotherapy were associated with a better survival rate. A total of 43.4% of patients relapsed, and their mean disease-free survival was 12.5 months. Conclusions: Primary ovarian leiomyosarcomas are more common in women in their 50s (mean age 53 years). Most of them are at an early stage at presentation. Advanced stage and mitotic count showed a detrimental effect on survival. Surgical excision associated with lymphadenectomy and chemotherapy are associated with increased survival. An international registry could help collect clear and reliable data to standardize the diagnosis and treatment.

## 1. Introduction

Primary ovarian sarcomas are rare and account for less than 2% of all ovarian malignancies [1]. Primary ovarian leiomyosarcoma (POLMS) is an extremely rare smooth muscle neoplasm that accounts for less than 0.1% of ovarian malignancies [2]. POLMS occurs mainly in women between the ages of 45 and 60 [2] and is usually a one-sided mass that grows to a large size [3]. POLMS is asymptomatic in the early stage; disease progression is rapid, and symptoms are non-specific in the advanced stage, with a high degree of malignancy [4]. Unfortunately, most patients present with advanced POLMS at diagnosis [4]. The most common symptoms are lower abdominal pain, altered bowel and bladder habits, vaginal bleeding, incontinence, or pain due to bilateral hydronephrosis [5,6]. Furthermore, tumor markers such as cancer antigen (CA) 125 are also ineffective because they are within the normal range or mildly elevated [7].

A leiomyosarcoma can arise in almost all organs other than neurogenic organs. Many leiomyosarcomas have been reported in the digestive tract, mesentery, uterus, vessels, retroperitoneum, and soft tissues [5]. Its macroscopic appearance is characterized by a solitary, lobular, soft fleshy solid mass with hemorrhage and cystic degeneration. Microscopically it presents 2/3 moderate/severe cytologic atypia, 10 + MF/10 HPF and tumor cell necrosis. POLMS may vary from well differentiated to highly pleomorphic sarcoma [8].

The pathogenesis of POLMS is unclear, since the ovary does not contain smooth muscle cells, though various hypotheses have been proposed to explain its origin. It may arise from smooth muscle cells of ovarian vessels or ligaments, or from remnants of Wolff’s duct, from transformation of totipotent ovarian mesenchymal cells or from muscle cells migrated from the uterus [9,10,11]. POLMS can also represent a malignant degeneration of a benign ovarian leiomyoma [10] or can arise within a benign cyst such as teratoma, serous cystadenoma or in papillary serous cystadenocarcinoma [11]. 

According to POLMS histogenesis, three histotypes can be identified: mesenchymal, teratoid and Müllerian. Different histotypes appear to occur in women of different ages and show different outcomes [7,10]. POLMS of mesenchymal origin usually occur in postmenopausal women and have an increased risk of metastasis, whereas those of teratoid origin occur in younger women and are usually unilateral [7,10].

No specific guidelines are available and the Gynecologic Cancer InterGroup (GCIG) has suggested following the recommendations for uterine leiomyosarcoma (ULMS) [12]. 

Surgical treatment is the cornerstone of management. It consists in radical resection of POLMS associated with abdominal hysterectomy, bilateral salpingo-oophorectomy and omentectomy [3]. POLMS is staged according to the classification system of the International Federation of Gynecology and Obstetrics (FIGO) as epithelial ovarian cancers (EOC) [9,10,11]. Stage, tumor size, mitotic index, grade and capsular invasion represent the most common prognostic factors [9,10,11]. Both chemotherapy and radiotherapy are used to prevent distant or local relapses, respectively, but there is no evidence of efficacy [9,13,14]. It is also not indicated what type of chemotherapy should be used [3]. Despite the combined treatments, POLMS remains an extremely lethal tumor with a 5-year survival rate less than 20% [14,15]. As it is a very rare tumor, it is not possible to identify clear guidelines regarding staging, treatment and prognosis. Recently, some literature reviews have tried to clarify the treatment and prognosis of patients with POLMS, though including only a part of the cases reported in the literature and analyzing only a part of the known potential prognostic factors [2,3,16]. Here, we reviewed the English literature since 1951, trying to obtain the largest possible sample of patients and analyzing all known prognostic factors, from clinical to histopathological, in order to identify the best treatment and clarify the outcome of POLMS patients.

## 2. Methods

### 2.1. Systematic Review of the Literature

We collected and analyzed the articles published in the English literature regarding POLMS from January 1951 to September 2022 using PubMed (https://pubmed.ncbi.nlm.nih.gov, accessed on 30 September 2022), Scopus (https://www.scopus.com/home, accessed on 30 September 2022), Web of Science (https://login.webofknowledge.com) research, and the terminologies “primary ovarian leiomyosarcoma”, “primary leiomyosarcoma of the ovary”, “primary sarcoma of the ovary”, “primary ovarian sarcoma” and “ovarian sarcoma”. We included all English papers describing POLMS such as interventional, observational, prospective and retrospective studies and case reports. Abstracts of medical conferences, editorials, preliminary studies with animal models and previous reviews were excluded. Papers reporting tumor of uncertain diagnosis, or the studies that had scant or aggregated data were excluded. Three authors [VM, AP, LC,] performed the literature review and collected data. Discrepancies were corrected in discussions with the principal investigator [VDM], and similarly correct data extraction was reviewed by the principal investigator [VDM]. A PRISMA (Preferred Reporting Items for Systematic Reviews and Meta-Analyses) flow chart with summary of search results is shown in Figure 1. 

We identified 152 articles on PubMed, 191 articles on Scopus and 113 articles on Web of Science databases. After duplicate exclusion, 231 records underwent first-step screening of titles and abstracts. Of these, 157 record were excluded because the tumor did not originate from the ovary or because the tumor was of uncertain diagnosis. A total of 74 full texts were considered for eligibility, and after reading them, 12 articles were excluded for being unfit according to the inclusion criteria or because they presented scant or aggregated data. An amount of 62 studies were finally included in the review, for a total of 113 POLMS patients (Table 1).

### 2.2. Statistical Analysis

Statistical analysis was performed using R Foundation for Statistical Computing (R-4.1.3), Vienna, Austria. Associations between clinical and pathological parameters were assessed by linear models and Fisher’s test. The non-parametric distribution of all the continuous variables was demonstrated by Shapiro test. Linear model results were expressed as odd ratio, 95% confidence interval (CI) and two-tailed *p*-value. The overall survival (OS) was computed as the time period from the date of surgery to either the date of death or last follow-up. Survival results were presented as hazard ratio (HR) and 95% confidence interval, and significance was expressed by log-rank *p*-value. Cox regression hazard model was used for multivariate analysis of the association between the clinical-pathological features and OS. Survival curves were plotted using the Kaplan–Meier method. Associations were considered statistically significant with a *p*-value lower than 0.05.

## 3. Results

### 3.1. Clinical Features

Table 2 shows the main clinical features of all 113 cases of POLMS reported in the literature. 

The age of the 113 affected patients at presentation ranged from 12 to 84 years (mean 53 years). Information about ethnicity was available in 109/113 (96.5%) patients. A total of 84/109 (77.1%) POLMS arose in Caucasian women, 22/109 (20.2%) in Asian women and 3/109 (2.7%) in African women.

Information about site origin was available in 63/113 (55.8%) patients. A total of 37/63 (58.7%) POLMS arose from the right ovary, 22/63 (34.9%) from the left ovary and 4/63 (6.3%) POLMS were bilateral. Symptoms were reported in 52/113 (46%) patients [1,4,5,6,9,10,13,16,19,20,21,25,26,27,28,31,32,33,34,35,37,38,39,40,41,44,46,47,48,49,51,52,53,54,55,56,57,58,59,60,61,62,63,64,66,67]. Each symptom occurred alone or associated with other symptoms. In particular, abdominal/pelvic pain was described in 38/52 (73.1%) patients [1,4,9,10,19,21,25,26,27,28,31,32,33,35,37,38,39,41,44,46,48,51,52,53,54,55,56,57,58,60,62,63,64,65,66,67], abdominal distension was described in 10/52 (19.2%) patients [6,10,15,16,20,44,56,57,62,65], anorexia weight loss was reported in 7/52 (13.5%) patients [6,10,15,16,20,44,56,57,62,65], urinary disorders were reported in 6/50 (12%) patients [4,19,27,47,53,61], weakness was reported in 5/52 (11.5%) patients [34,35,37,46,65], constipation was reported in 3/52 (5.8%) [5,28,38], fever was reported in 3/50 (6%) patients [37,46,65], rectal hemorrhage was reported in 1/52 (1.9%) patients [65], metrorrhagia was reported in 2/52 (3.8%) patients [33,63] and vaginal prolapse was reported in 1/50 (2%) patients [67]. A total of 3/52 (5.8%) patients were asymptomatic [15,40,59].

### 3.2. Surgery

Information about POLMS treatment was reported in 80/113 (70.8%) patients [1,2,4,5,6,9,10,15,16,19,20,21,22,25,26,27,28,29,30,31,32,33,35,36,37,38,39,40,41,42,43,44,45,46,47,48,49,50,51,52,53,54,55,56,57,58,59,60,61,62,63,64,65,66,67]. POLMS was treated by surgical resection in 76/80 (95%) patients [1,2,4,5,6,9,10,15,16,19,20,22,25,26,27,28,29,30,31,32,33,35,36,37,38,39,40,41,42,43,44,45,46,47,48,49,50,51,52,53,54,55,56,57,58,59,60,61,62,63,64,65,66,67]. A total of 34/80 (42.5%) patients received surgery without lymphadenectomy [2,9,13,14,16,19,20,22,26,28,29,30,34,35,36,37,39,40,43,57,58,59,60,64,65], 10/80 (12.5%) of patients received surgery with lymphadenectomy [1,4,5,38,47,48,51,61,62,65], 21/80 (26.3%) patients received surgery without lymphadenectomy associated with chemotherapy [2,25,27,28,30,31,32,33,41,45,46,49,50,54,55,63,67] and 11/80 (13.8%) patients received surgery with lymphadenectomy associated with chemotherapy [6,10,15,42,52,53,56,65,66]. A total of 3/80 (3.8) POLMS patients did not receive treatment [2,24,26]. A total of 1/80 (1.2%) POLMS patients received only chemotherapy [2].

### 3.3. Adjuvant Treatment

Information about adjuvant treatment was reported in 78/113 (69%) patients [1,2,4,5,6,10,15,15,16,16,17,18,19,19,20,20,21,21,22,22,23,24,25,26,27,28,29,30,31,32,33,35,36,37,38,39,40,41,42,44,45,46,47,48,49,50,51,52,53,54,55,56,57,58,59,60,61,62,63,64,65,66,67], 39/78 (50%) received adjuvant treatment [2,6,10,15,20,22,27,29,30,31,32,33,34,35,37,42,45,46,49,50,52,53,54,55,56,63,65,66,67], 31/39 (79.5%) patients received chemotherapy [2,6,10,15,27,30,31,32,33,41,42,44,45,46,50,52,53,54,55,56,63,66,67], 2/39 (5.1%) patients received chemotherapy associated with radiotherapy [25,49], 5/39 (12.8%) patients received radiotherapy [20,22,29,35,37] and 1/39 (2.6%) patients received hormone therapy [66]. Information about the type of chemotherapy used was available in 24/33 (72.7%) patients [2,6,10,15,27,31,33,41,46,52,53,56,61,63,65,66,67], 17/24 (70.8%) patients received a schedule for sarcoma [6,10,15,27,33,46,52,53,56,65,66,67], 5/24 (20.8%) received a schedule for sarcoma associated with platinum [31,32,33,41,63] and 2/24 (8.4) received a schedule as for dysgerminoma [50]. 

### 3.4. Risk Factors

The tumor size was available in 73/113 (64.6%) patients [1,5,6,10,13,15,16,19,20,21,22,25,26,27,28,31,34,36,37,38,39,40,41,44,46,48,51,52,53,54,55,56,57,58,59,60,61,62,63,64,65], ranging from 33 to 350 mm, with a mean size of 151.2 mm (SD +/− 68.5). CA 125 was reported in 32/113 (28.3%) patients [1,4,5,6,10,13,14,15,16,34,38,40,41,47,48,49,50,51,52,53,54,55,56,57,59,61,62,63,65,66,67], 19/32 (59.4%) patients had a normal value [4,5,6,14,15,34,38,40,47,49,52,53,56,57,59,61,62,65] and 13/32 (40.6%) patients had an elevated value [1,10,13,16,41,48,54,55,63,67]. Carcinoembryonic antigen (CEA) was reported in 16/113 (14.1%) patients [5,10,27,28,29,30,34,47,49,50,53,57,58,59,63,64], 15/16 (93.8%) patients had a normal value [5,13,14,15,34,38,41,48,49,61,66,67] and 1/16 (6.2%) had an elevated value [59]. CA 19–9 was reported in 10/113 (8.8%) patients, 9/10 (90%) patients had a normal value [1,5,10,15,38,41,48,59,61] and 1/10 (10%) patients had an elevated value [67]. Human epididymis protein (HE4) was assessed in only one patient and resulted elevated [16]. The stage was available in 89/113 (78.8%) patients. A total of 52/89 (58.4%) patients were stage I, 8/89 (9%) patients were stage II, 25/89 (28%) patients were stage III, and 4/89 (4.6%) patients were stage IV. Mitotic count was reported in 81/113 (71.7%) patients [1,4,5,6,9,13,14,15,16,18,19,20,22,23,25,26,28,29,31,32,33,34,35,36,37,38,39,40,41,43,44,45,46,47,49,51,52,53,54,55,56,57,58,59,60,61,62,63,64,65], and the mean number of mitoses was 14.8 (range 1–80).

### 3.5. Immunohistochemistry

Vimentin examination was available in 24/113 (21.2%) patients [1,6,9,10,14,16,32,33,41,44,45,47,48,49,51,53,54,55,57,58,61,62] and was positive in all patients (100%). Smooth muscle actin (SMA) examination was available in 37/113 (32.7%) patients [1,4,5,6,9,10,13,14,15,16,30,33,34,38,40,41,45,46,48,49,50,51,53,54,55,56,57,58,59,60,61,62,64,67]; 35/37 (94.6%) patients were positive [1,4,5,6,9,10,13,14,15,16,32,33,34,38,40,41,45,47,48,49,51,53,54,55,56,57,58,59,60,62,64,67] and 2/37 (5.4%) were negative [46,61]. Desmin examination was available in 28/113 (24.8%) patients [4,5,6,13,14,15,16,31,32,34,38,44,45,46,47,48,51,53,54,55,56,59,60,61,62,64,67]; 24/28 (85.7%) were positive [4,5,13,14,15,16,31,32,34,38,45,46,47,48,51,53,54,55,56,59,60,64] and 4/28 (14.3%) were negative [6,61,62,67]. S100 examination was available in 11/113 (9.7%) patients [13,15,16,38,45,46,51,60,61,62]; 3/11 (27.3%) were positive [16,51,61] and 8/11 (72.7%) were negative [13,15,38,45,46,60,62]. CD34 testing was available in 13/113 (11.5%) patients [4,10,15,16,20,46,49,52,59,60,61,62,67] and was positive in 2/13 (15.4%) [4,16] and negative in 11/13 (84.6%) patients [10,15,46,49,52,59,60,61,62,67]. CD68 testing was available in 3/113 (2.6%) patients [10,16,46] and was positive in 2/3 (66.7%) patients [16,27] and negative in 1/3 (33.3%) patients [10]. Ki-67 testing was available in 12/113 (10.6%) patients [1,5,14,16,38,45,51,53,60,62,64,67] and was negative in 2/12 (16.7%) patients [14,64] and positive in 10/12 (83.3%) patients [1,5,16,38,45,51,53,60,62,67], with mean value of 26.3% and range between 10% and 50%.

### 3.6. Follow-Up Data

Follow-up information was available for 100/113 (88.5%) patients (Table 2) [1,2,4,5,6,10,13,14,15,16,17,18,19,21,22,23,24,25,26,27,28,29,30,31,32,33,35,36,38,39,40,41,42,43,44,45,46,47,48,49,50,51,52,53,54,55,56,57,58,59,60,61,62,63,65,66]. In total, 2/100 (2%) patients died for causes other than POLMS [36,41] and 13/100 (13%) were lost in follow-up [9,20,34,37,43,49,64,67]. A total of 46/100 (46%) patients were alive and disease-free [1,2,4,6,10,15,16,22,24,25,26,27,28,31,32,33,39,40,42,43,45,47,51,52,53,55,56,57,58,59,60,62,63,65,66], 48/100 (48%) patients died of disease (DOD) [2,5,13,14,17,18,19,21,23,29,30,33,35,38,43,44,46,48,50,61,65] and 4/100 (4%) patients were alive with disease (AWD) [2,54]. Follow-up time was available for 99/113 (87.6%) patients [1,2,4,5,6,10,14,15,16,17,18,19,22,23,24,27,29,30,32,35,36,38,42,45,46,47,48,50,53,54,55,59,61,62,63,65,66]; mean follow-up was 26 months (range 1–144). In total, 56/99 (56.6%) patients did not relapse [1,2,4,10,13,15,21,22,23,24,25,26,27,28,29,32,33,36,39,40,41,42,43,45,47,48,50,51,52,53,54,55,56,57,58,59,62,65,66], 43/99 (43.4%) of patients relapsed [2,5,6,14,16,17,18,19,23,30,31,33,35,37,38,39,43,44,46,60,61,63,65] and their mean disease-free survival (DFS) was 12.5 months (range 1–67).

In 32/44 (72.7%) patients, the site of recurrence was reported [5,6,14,16,17,18,19,23,30,31,33,35,38,39,43,44,46,60,61,63,65], including 15/32 (46.9%) in the pelvis [14,16,17,18,19,23,30,31,33,35,38,39,43,44,46,60,61,63,65], 9/32 (28.1%) in the upper abdomen [5,6,19,23,43,44,46,60], 8/32 (25%) in the thorax [14,30,35,43,44,63], 2/32 (6.2%) with metastasis above and below the diaphragm [14,44], 1/32 (3.1%) in the retroperitoneum [38], and 1/32 (3.1%) in the preauricular lymph node [43]. Management of recurrence was available for 19/43 (44.2%) patients [2,6,14,16,27,35,44,60,63,65], including 3/19 (15.8%) who received cytoreduction surgery [6,14,16], 2/3 (66.7%) of whom also received chemotherapy [14,16]; 6/19 (31.6%) did not received treatment [2,19,35,44,65] and 10/19 (52.6%) received only chemotherapy [2,60,63].

The survival analysis showed a significant difference in OS between patients with stage I–II (median OS: 54 months) and patients with stage III–IV (median OS: 18 months) (HR = 4.2, 95%CI = 2.1–8.4, *p* < 0.0001) (Figure 2A), as well as reduced OS in patients with a mitotic count >10 (median OS: 35 months) in comparison with those with a lower number of mitoses (median OS: 63 months) (HR = 3.0, 95%CI = 1.3–7.1, *p* = 0.0097) (Figure 2B).

The analysis of the OS curves of patients subjected to different treatments showed that surgery improved the prognosis of POLMS (Figure 3A)(Surgery: HR = 0.27, 95% CI = 0.08–0.96, *p* = 0.042, Surgery + CHT: HR = 0.21, 95% CI = 0.05–0.80, *p* = 0.022); however, the choice of treatment was dependent on the tumor stage (Figure 3B) (*p* = 0.007), while no significant association was observed with mitotic count (Figure 3C).

Focusing on the patients who underwent surgery, we registered a significantly different effect of treatments on OS (*p* = 0.016, Figure 4A) and risk of death (*p* = 0.032, Figure 4B). In particular, patients treated with surgery including lymphadenectomy combined with chemotherapy showed a better prognosis, and no events of death were registered during follow-up. In this case, the choice of treatment was dependent on the tumor stage (*p* = 0.026, Figure 4C) but was also influenced by the mitotic count (*p* = 0.046, Figure 4D). In fact, both these variables were significantly associated with the risk of death in POLMS patients (Figure 4E,F).

A multivariate analysis confirmed that the stage was the principal prediction factor for the risk of death in these patients, independently from the treatment choice. (Table 3).

Subsequently, we considered only patients treated with chemotherapy and compared the OS of patients surgically treated with or without lymphadenectomy (Figure 5A). Interestingly, a very significant difference (*p* = 0.0011) in OS was observed, confirming the efficacy of lymphadenectomy in improving the prognosis of POLMS patients. Similarly, the comparison between patients who underwent surgery with lymphadenectomy in combination or not with chemotherapy (Figure 5C) confirmed the significantly improved OS in patients who received adjuvant treatment (*p* = 0.0045). Interestingly, in these subgroups of patients, the stage did not influence the choice of the most suitable treatment (Figure 5B,D).

## 4. Discussions

To our knowledge, this study is the literature review that includes the largest number of POLMS patients [2,3,65]. Since 1951, when the first case of POLMS was described by Istre, only 113 cases of POLMS have been reported in the literature [1,2,4,5,6,9,10,15,16,17,18,19,20,21,22,23,24,25,26,27,28,29,30,31,32,33,35,36,37,38,39,40,41,42,43,44,45,46,47,48,49,50,51,52,53,54,55,56,57,58,59,60,61,62,63,64,65,66,67].

Our review highlights the uncertainties in the treatment of POLMS. In fact, although the guidelines recommend treating POLMS as a ULMS, most POLMSs are treated as EOCs. It should be kept in mind, however, that there are many differences between the two types of tumors despite the same site of onset. Our review also highlighted how the stage, the number of mitoses and the type of treatment are factors that can influence the prognosis of POLMS. Although POLMS tumors are treated according to EOC guidelines, several differences have emerged between these two ovarian cancers. EOCs are usually diagnosed at an advanced stage in women with a median age of 63 years [67]. In our review, women affected by POLMS had a median age of 53 years old (range: 12–84 years, SD +/− 17.6) and were for the most part Caucasian (77%, 84/109). Different from a previous study [4], POLMS was found at early stage in 67.4% (60/89) of cases. Usually, this type of tumor presents as a unilateral (93.6%, 59/63) bulky mass (15.1 cm, range 3.3–35 cm, SD +/− 6.9 cm), generally at the right ovary. The most common symptom is abdominal/pelvic pain followed by other symptoms due to the space-occupying mass such as abdominal distension, anorexia, weight loss, constipation and urinary disorders. Rarely, rectal [65] or vaginal bleeding [35,44] is also described and asymptomatic patients are anecdotal [15,40]. Unlike EOC, POLMS patients usually present normal or slightly increased tumor markers. CA 125 was reported in less than a third of patients [1,4,5,6,10,13,14,15,16,34,38,40,41,47,48,49,50,51,52,53,54,55,56,57,59,61,62,63,65,66,67], mostly in the normal range and found to be elevated in 40% of patients, increasing up to 202 IU [1,10,13,16,41,48,54,55,63,67]. CA 125 is a high molecular weight mucinous glycoprotein found in adult tissues derived from the coelomic and Mullerian epithelia. CA 125 is expressed in different tissues such as ovary, endocervix, endometrium, pleura, pericardium, peritoneum, secretory mammary glands, apocrine sweat glands, intestines, lungs and kidneys. CA 125 shows elevated level in several gynecological malignancies particularly in 50% of early-stage EOC and in 92% of advanced-stage EOC [68]. Moreover, elevated CA 125 levels may indicate advanced non-cytoreducible EOC, persistence or recurrence of EOC [68]. In contrast, the predictive role of CA 125 levels has not been conclusively demonstrated in ULMS. A recent study showed that ULMS patients with a higher CA 125 at diagnosis tended to recur more [69,70]. Similarly, HE4 and CEA have been used for the diagnosis, cytoreducibility and recurrence risk of ovarian cancers [71]. Moreover, HE4 concentration was higher also in patients with ULMS and has been proposed to distinguish ULMS from leiomyomas [72]. However, the dosage of HE4 and CEA in the POLMS patients included in our review was only anecdotal; therefore, it was not possible to analyze any correlations between the tumor markers and the POLMS.

Preoperative POLMS diagnosis is difficult because ultrasound may be ambiguous, showing both benign and malignant features [14] (Figure 6A,B).

A recent report described imaging findings of a POLM correlated with histopathologic features [64]. Contrast-enhanced computed tomography (CT) showed a pelvic multilocular mass with heterogeneous enhancement due to colliquative intralesional areas surrounded by solid peripheral components (Figure 7A–C).

Coronal image reconstruction with maximum intensity projection (MIP) was useful to evaluate the vascular origin of the mass by studying the complete course of the vessel [64]. Magnetic resonance imaging (MRI) showed an isointense mass to muscle on T1-weighted images and heterogeneously hyperintense on T2-weighted images, surrounded by a subtle perilesional fluid. Diffusion-weighted images (b = 800 s/mm^2^) and apparent diffusion coefficient (ADC) map revealed restriction of the diffusion with minimum ADC value of 0.81 × 10^−3^ mm^2^/s and mean ADC value of 1.24 × 10^−3^ mm^2^/s. 

Histological diagnosis of ovarian leiomyosarcoma can also be complex [45]. This is based on knowledge of ULMS [73]. POLMS diagnosis is made when at least two of the three diagnostic criteria (coagulative necrosis, cellular atypia and mitotic index > 10 per high-power fields (HPF)) are present (Figure 8A) [20]. However, POLMS may also be diagnosed when the mitotic index is <10 per HPF and there is no necrosis but pronounced atypia is present [15,45,46] (Figure 8B).

In our review there were 22 cases of POLMS with mitotic index < 10/HPF [1,6,9,15,22,29,32,33,39,41,43,45,53,60] and the mean number of mitoses was 14.6 (range 1–80). 

According to the findings of Yuksel et al. [65], POLMS patients with a mitotic count > 10 had a worse prognosis than patients with fewer mitoses (median OS: 35 vs. 63 months, respectively, *p* = 0.0097) (Figure 2). Commonly, mitotic count is evaluated by pathologists in making a diagnosis of cancer, and to grade malignancy, informing prognosis [74]. Mitosis is a process of cell cycle in which replicated chromosomes are divided into two new nuclei producing genetically identical cells retaining their chromosomes number [75]. Mitotic count is an indicator of the cell proliferation rate and hence the aggressiveness of several cancers such as breast cancers, ULMS, astrocytoma, gastrointestinal stromal tumours, etc. [76,77,78].

POLMS should be distinguished from fibrosarcomas, rhabdomyosarcomas, thecomas and extradigestive stromal tumors. Immunohistochemistry may be useful in diagnosing POLMS and distinguishing its subtype, and thus the biological behavior and subsequent prognosis. POLMSs are typically positive for actin, vimentin [34] and desmin; S-100, caldesmon are other common markers found in them [10,55,59]. POLMS of vascular origin show positivity for h-caldesmon and focally positivity or negativity for desmin, while those of non-vascular origin are negative for h-caldesmon with variable levels of desmin expression [13]. Unlike recent reviews [2,3,65], we investigated POLMS’ immunohistochemical markers. In our review, the most commonly investigated markers were SMA (32.7% of POLMS patients) [1,4,5,6,9,10,13,14,15,16,30,33,34,38,40,41,45,46,48,49,50,51,53,54,55,56,57,58,59,60,61,62,64,67], desmin (24.8% of POLMS patients) [4,5,6,13,14,15,16,31,32,34,38,44,45,46,47,48,51,53,54,55,56,59,60,61,62,64,67], and vimentin (21.2% of POLMS patients) [1,6,9,10,14,16,32,33,41,44,45,47,48,49,51,53,54,55,57,58,61,62]. The most frequently positive markers were vimentin (100% of cases) [1,6,9,10,14,16,32,33,41,44,45,47,48,49,51,53,54,55,57,58,61,62], SMA (94.6% of patients) [1,4,5,6,9,10,13,14,15,16,32,33,34,38,40,41,45,47,48,49,51,53,54,55,56,57,58,59,60,62,64,67] (Figure 8D), and desmin (85.7% of cases) [4,5,13,14,15,16,31,32,34,38,45,46,47,48,51,53,54,55,56,59,60,64]. 

However, it was not possible distinguish between POLMS of mesenchymal origin from those of teratoid origin thus correlating their origin with prognosis.

Ki-67 was rarely reported in the literature (10.6% of POLMS) [1,5,14,16,38,45,51,53,60,62,64,67] but frequently expressed if investigated (83.3% of cases) [1,5,16,38,45,51,53,60,62,67].

Ki-67 is a nuclear marker closely related to tumor cell proliferation and growth (Figure 9). It correlates with tumor stage and metastasis, and its expression is significantly higher in tumors with poorly differentiated cells [48]. In a previous paper, Mayerhofer reported a Ki-67 expression of 30% [41]; in our review, Ki-67 expression ranged between 10% and 50% with a mean value of 26.3%. In the aforementioned study, Mayerhofer et al. [41] found POLMS positivity for matrix metalloproteinases MMP1 and MMP2, which may be linked to tumor aggressiveness. Bodner et al. found a high tumor positivity to B-cell lymphoma/leukemia-2 gene (BCL2) [12]. Members of the BCL-2 protein family play an important role in the control of apoptosis, an overexpression of pro-survival BCL-2 proteins or a reduction of pro-apoptotic BCL-2 proteins, both resulting in inhibition of apoptosis, with ensuing increased cell replication even in the absence of growth factors [79]. However, in our review we were unable to find a correlation between immunohistochemical markers and POLMS prognosis. 

The GCIG consensus recommends total hysterectomy and bilateral salpingo-oophorectomy in POLMS limited to the ovary, and for patients who did not undergo lymph node dissection or omentectomy, a second operation is not considered necessary because of the low risk of occult metastasis [13]. In our review, 95% (76/80) of patients underwent surgery [1,2,4,5,6,9,10,15,16,19,20,22,25,26,27,28,29,30,31,32,33,35,36,37,38,39,40,41,42,43,44,45,46,47,48,49,50,51,52,53,54,55,56,57,58,59,60,61,62,63,64,65,66,67] and 12.5% (10/80) of these patients also underwent lymphadenectomy [1,4,5,38,47,48,51,61,62,65] and 13.8% (11/80) received chemotherapy after lymphadenectomy [6,10,15,42,52,53,56,65,66]. Most patients (17/24, 70.8%) were given the sarcoma regimen [6,10,15,27,33,46,52,53,56,65,66,67] and platinum (cis and carboplatin) was added in only 20.8% (5/24) of cases [31,32,33,41,63]. In our review, tumor stage was the principal factor affecting survival. As expected, [65,80,81,82,83,84] patients with early stage (I–II) POLMS showed significantly better OS than patients with advanced stage (III–IV) POLMS (median OS: 54 vs. 18 months, respectively, *p* < 0.0001) (Figure 2A). Surgery also improved the prognosis of POLMS (Figure 3A), although the choice of treatment strongly depended on the stage of the tumor (Figure 3B) (*p* = 0.007). Moreover, different treatments were associated with different OS rates. Notably, patients treated with surgery including lymphadenectomy combined with chemotherapy showed a better prognosis and no deaths were recorded during follow-up. Particularly, in these patients, the choice of the treatment was dependent on the tumor stage (*p* = 0.026, Figure 4C) but was also influenced by the mitotic count (*p* = 0.046, Figure 4D). Both these variables resulted significantly associated to the risk of death in POLMS patients (Figure 4E,F).

According to GCIG consensus, routine use of chemotherapy is not recommended in POLMS limited to the ovary [12]. Nevertheless, in our review 19.2% (10/52) of POLMS patients at first stage received chemotherapy [10,15,25,27,32,45,53,55,65] and in 1 patient radiotherapy was also administered [22]. Considering only patients treated with chemotherapy and comparing the OS of patients surgically treated with or without lymphadenectomy (Figure 5A), we observed a very significant difference (*p* = 0.0011) in OS, confirming the efficacy of lymphadenectomy in improving the prognosis of POLMS patients. As suggested in a recent paper [65], lymphadenectomy may improve DFS. Differently from this recent review [65], we did not analyze pelvic lymph node dissection separately from lumbo-aortic lymph node dissection, but we obtained similar findings. In our analysis, the comparison between patients who underwent surgery with lymphadenectomy in combination or not with chemotherapy (Figure 5A) confirmed the significantly improved OS in patients who received it (*p* = 0.0045). Notably, in these subgroups of patients, the stage did not influence the choice of the most suitable treatment (Figure 5B,D). 

In our review, 46% (46/100) of patients were disease-free [1,2,4,6,10,15,16,22,24,25,26,27,28,31,32,33,39,40,42,43,45,47,51,52,53,55,56,57,58,59,60,62,63,65,66]; 48% (48/100) patients were DOD [2,5,13,14,17,18,19,21,23,29,30,33,35,38,43,44,46,48,50,61,65], and 4% (4/100) of patients were AWD [2,54]. Follow-up time was only available for 87.6% (99/113) of patients [1,2,4,5,6,10,14,15,16,17,18,19,22,23,24,27,29,30,32,35,36,38,42,45,46,47,48,50,53,54,55,59,61,62,63,65,66], and it was too short to record all the deaths (mean follow-up was 26 months, range 1–144). However, 56.6% (56/99) of patients did not relapse [1,2,4,10,13,15,21,22,23,24,25,26,27,28,29,32,33,36,39,40,41,42,43,45,47,48,50,51,52,53,54,55,56,57,58,59,62,65,66], 43.4% (43/99) of patients relapsed [2,5,6,14,16,17,18,19,23,30,31,33,35,37,38,39,43,44,46,60,61,63,65] and their mean DFS was 12.5 months (range 1–67). On the other hand, a recent case series reported that 26% of patients were free of disease (FOD) with DFS of 16 months [2]. 

The most common sites of recurrence were pelvis (46.9%) [14,16,17,18,19,23,30,31,33,35,38,39,43,44,46,60,61,63,65], upper abdomen (28.1%) [5,6,19,23,43,44,46,60] and thorax (25%) [14,30,35,43,44,63]. Most of the relapsed patients received only palliative care [2,60,63], and only 15.8% of the relapse patients received cytoreduction surgery [6,14,16].

It is well known that EOC should be treated in high-volume referral centers both to ensure optimal treatment with a multidisciplinary team and to ensure the best possible prognosis [85,86,87]. Since POLMS is an even rarer tumor, POLMS should necessarily be centralized both to guarantee the best possible treatment and to build databases useful for dissolving the doubts that still remain on the management of this tumor.

Although our review includes cases collected over an extensive period of time (70 years) with the risk of including cases with no verified diagnosis and subjected to non-homogeneous treatments, unfortunately with a very rare tumor such as POLM, such a long-term review would seem, to date, the only solution to obtain indications on management in the absence of better-quality data. Including as many patients as possible and analyzing as many diagnostic and therapeutic features as possible is an attempt to increase knowledge about POLMS treatment. 

## 5. Conclusions

POLMS is a very rare neoplasm, and very few case reports and case series are available in the English literature. Our review underlined the difference between POLMS and EOC (age, unilaterality of lesion, increased aggressiveness, little or no increase in tumor markers) but did not resolve the dilemma about the surgical approach, i.e., should we treat POLMS as a ULMS or as an EOC? Should we perform staging lymph node dissection as in early stage EOC or should we only remove enlarged lymph nodes as in advanced EOC and ULMS? Moreover, we confirmed that advanced stage and high number of mitoses have a negative effect on survival and that the type of treatment may also influence survival. Both lymphadenectomy and chemotherapy could improve survival of patients with POLMS. However, even today there are too few data in the literature to identify the right diagnosis and to clarify the treatment of POLMS. Only an international POLMS registry could help collect clear and reliable data to standardize the diagnosis and treatment of this extremely rare and aggressive cancer.

## Figures and Tables

**Figure 1 cancers-15-02953-f001:**
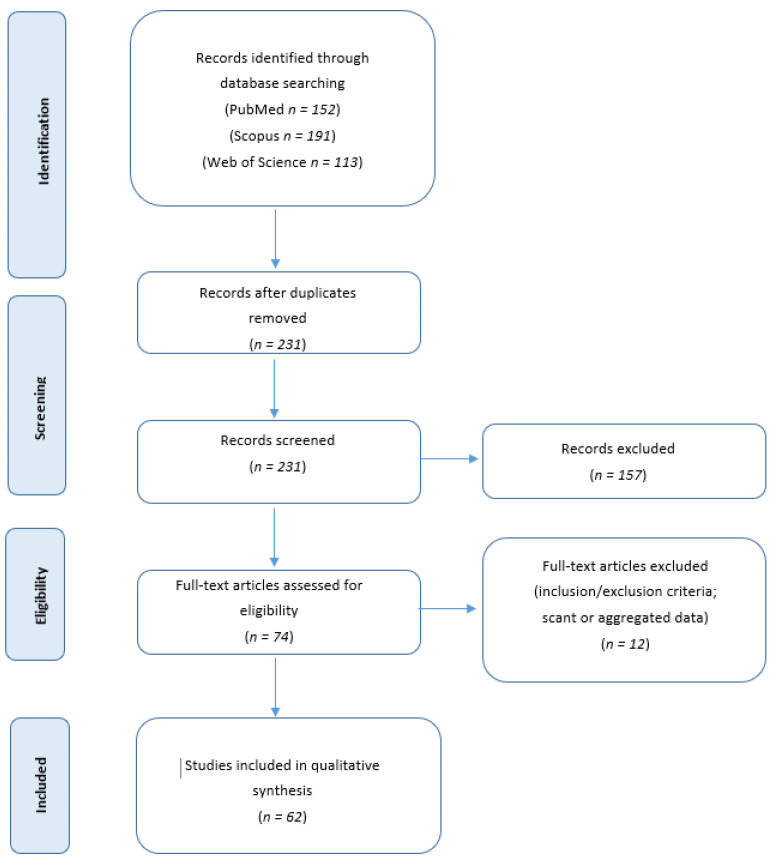
Review of the literature: Preferred Reporting Items for Systematic Reviews and Meta-Analyses (PRISMA) flow chart.

**Figure 2 cancers-15-02953-f002:**
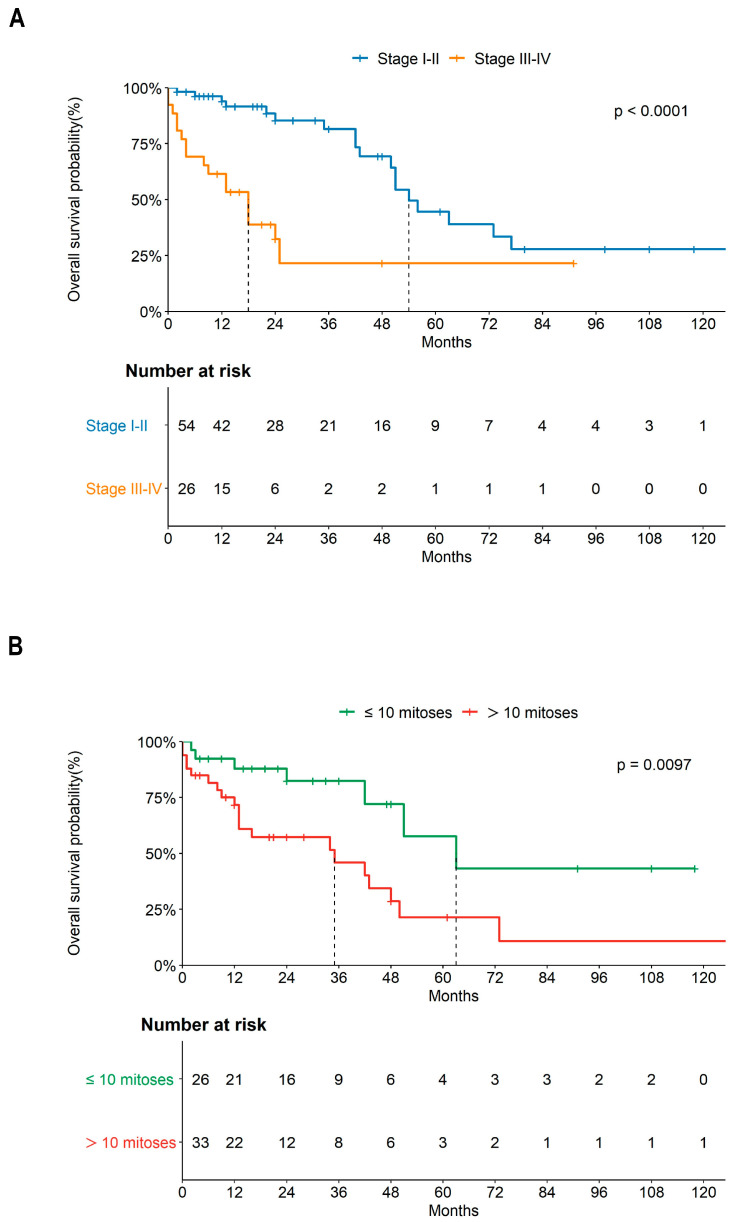
Impact of stage (**A**) and mitotic count (**B**) on OS of patients with POLMS.

**Figure 3 cancers-15-02953-f003:**
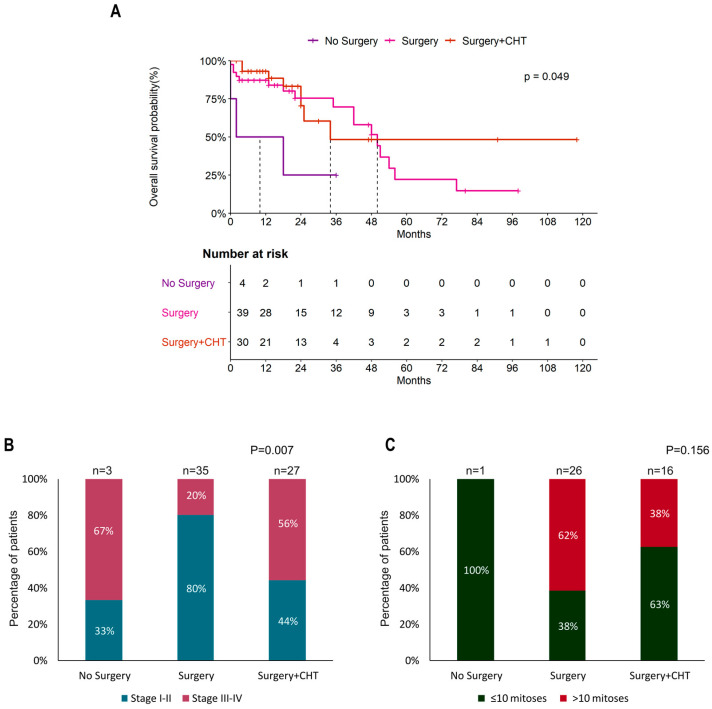
Impact of surgery on OS of patients with POLMS (**A**). Impact of tumor stage on the choice of the treatment of POLMS patients (**B**). Impact of mitotic count on the choice of the treatment of POLMS patients (**C**).

**Figure 4 cancers-15-02953-f004:**
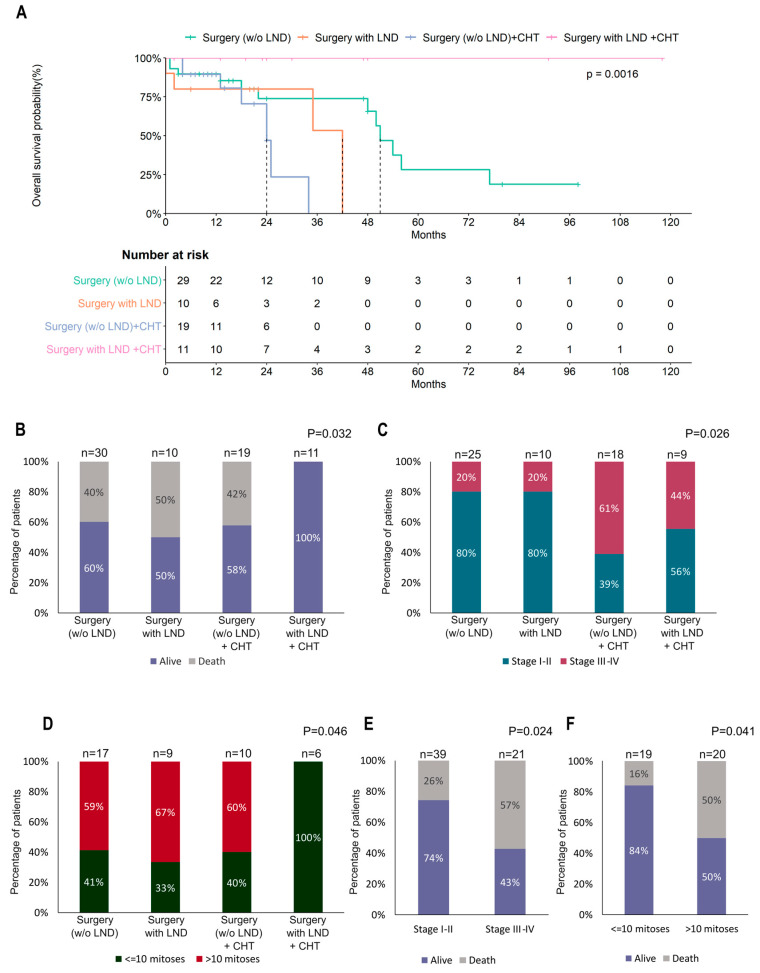
Impact of the different treatments on OS (*p* = 0.016, Figure 4A) and on the risk of death (*p* = 0.032, Figure 4B) of POLMS patients. Impact of stage (**C**) and mitotic count (**D**) on the choice of treatment of POLMS patients. Impact of stage (**E**) and mitotic count (**F**) on the risk of death in POLMS patients.

**Figure 5 cancers-15-02953-f005:**
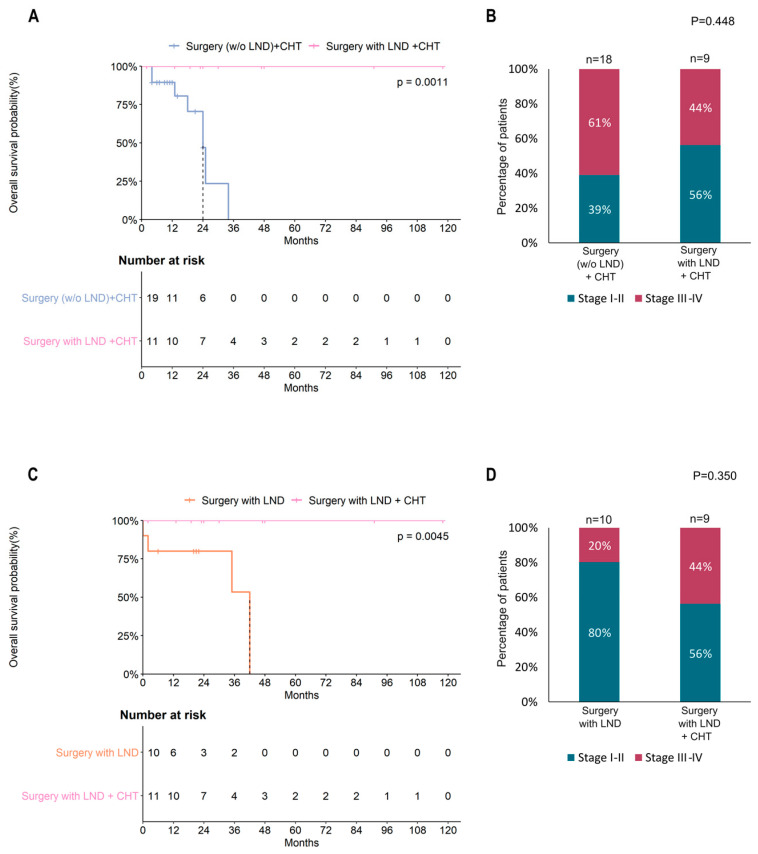
Impact of lymphadenectomy on OS of POLMS patients treated with chemotherapy (**A**). Impact of chemotherapy on OS of POLMS patients treated with lymphadenectomy (**C**). Impact of stage on the choice of chemotherapy (**B**) and lymphadenectomy (**D**).

**Figure 6 cancers-15-02953-f006:**
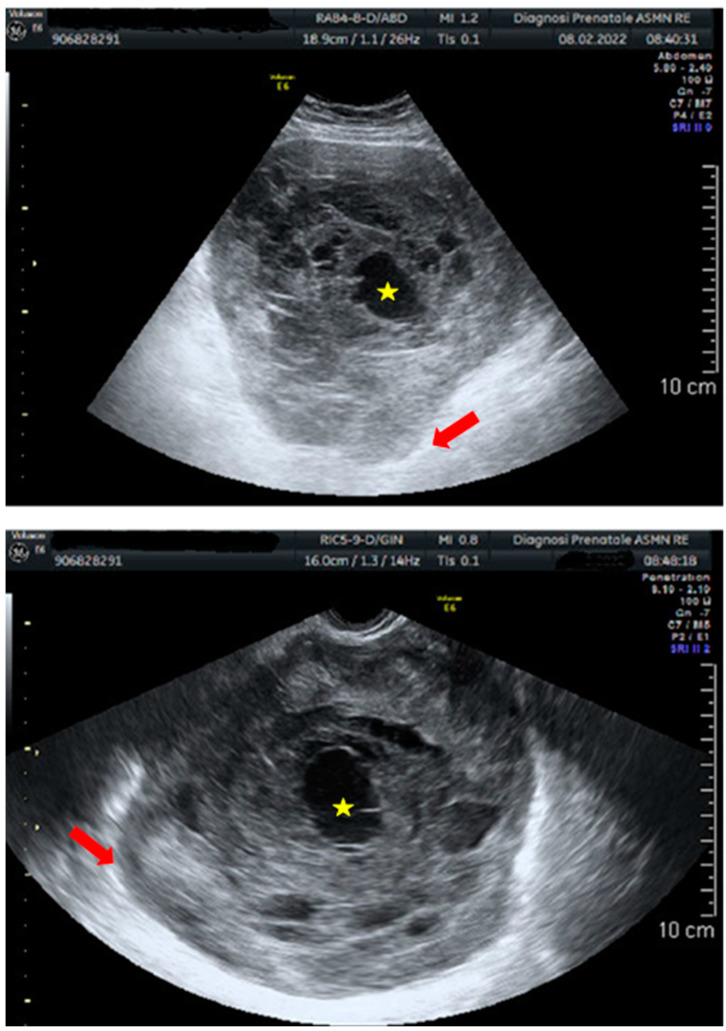
Ultrasound appearance of POLMS: a huge pelvic mass of 20 × 15 × 16 cm occupying the pelvis going beyond the transverse umbilical, it appears as a solid mass with irregular margins (red arrow), dishomogeneous echostructure due to the presence of anechoic, poorly vascularized cystic areas at color doppler (yellow star) (previously unpublished, original photos).

**Figure 7 cancers-15-02953-f007:**
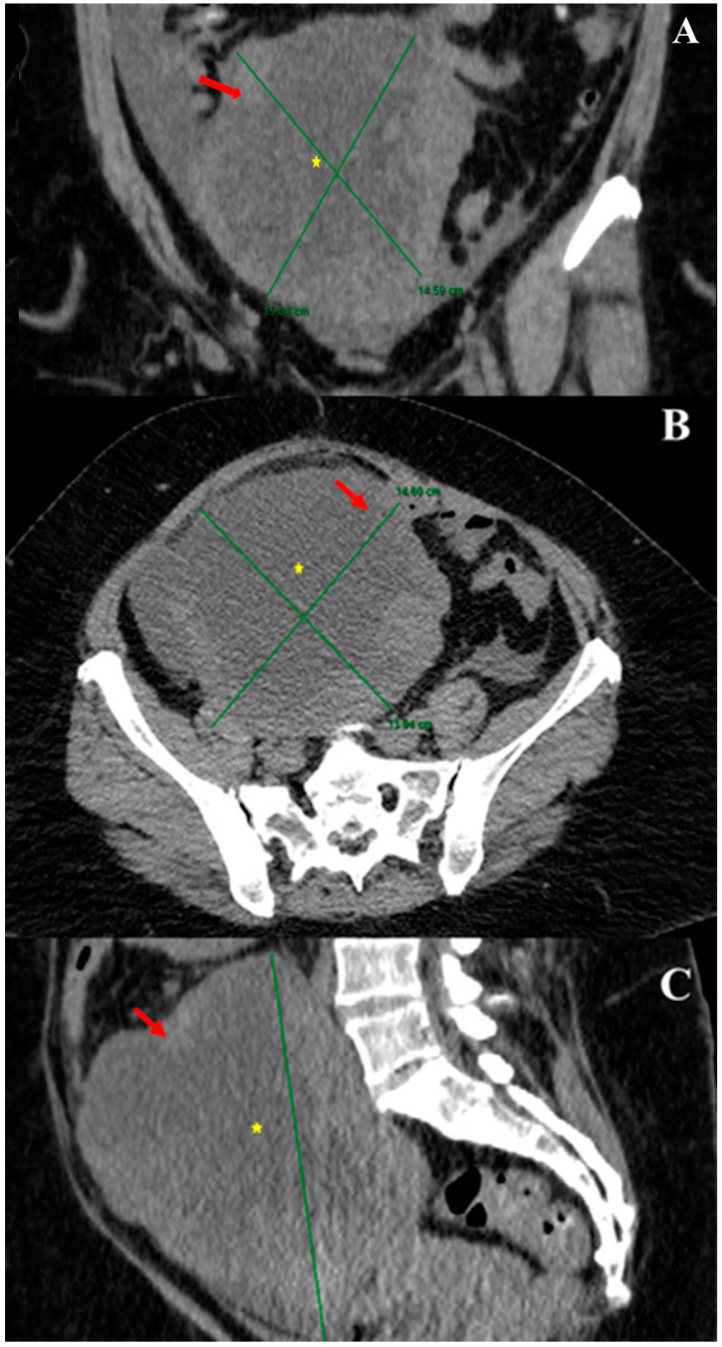
Computed tomography appearance of POLMS: Coronal (**A**), Axial (**B**) and Sagittal (**C**) plane: gross expansive lesion originating from the right adnexa adhering to the right wall of the uterus. It presents a mixed-cystic solid structure with a central fluid component (yellow star) and multiple solid peripheral vegetations (red arrow) with contrastographic enhancement (previously unpublished, original photos).

**Figure 8 cancers-15-02953-f008:**
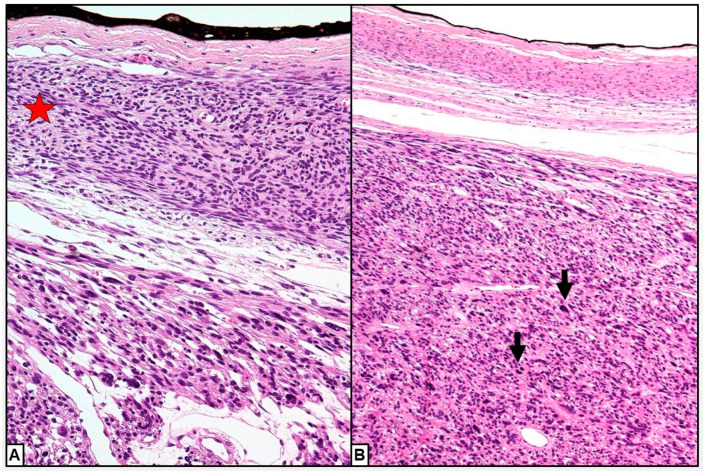
The tumor revealed a proliferation of spindle cells with hyperchromasia and moderate-severe nuclear atypia. The tumor borders were predominantly pushing with a residual peripheral focus of ovarian parenchyma (1A, star). Mitotic figures were frequently identified (1B, arrows) (**A**): Hematoxylin and eosin, 20 ×; (**B**): Hematoxylin and eosin, 10 ×)) (previously unpublished, original photos).

**Figure 9 cancers-15-02953-f009:**
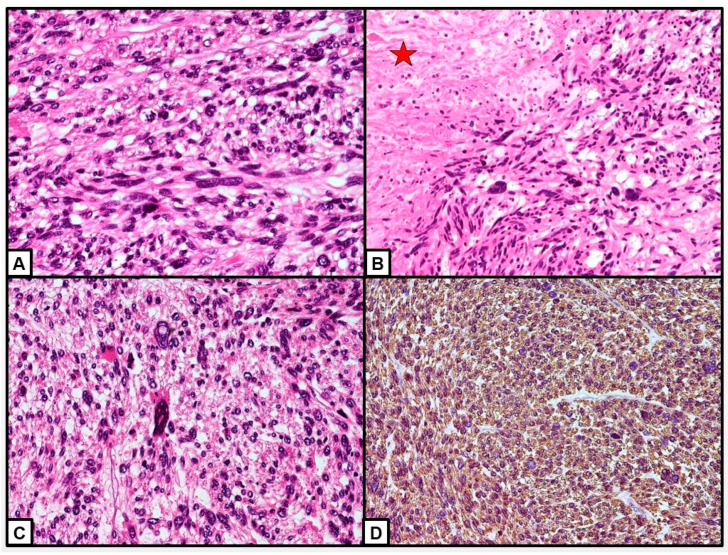
Histological details of the tumor cells. Spindle cell morphology. Severe pleomorphism. Necrotic areas (2B, star) (hematoxylin and eosin; (**A**): 40 ×; (**B**): 20 ×; (**C**): 40 ×). (**D**) On immunohistochemical exam, the tumor cells were positive for smooth muscle actin (20 ×) (previously unpublished, original photos).

**Table 1 cancers-15-02953-t001:** Clinical features of 113 primary ovarian leiomyosarcomas reported in the literature.

FIRST AUTHOR	YEAR	AGE	ETHNICITY	SYMPTOMS	SITE	SIZE (mm)	CA 125	TREATMENT	ADJUVANT THERAPY	STAGE	RECURRENCE	TIME RECURRENCE	TREATMENT RECURRENCE	THERAPY RECURRENCE	FOLLOW-UP (months from first diagnosis)	STATE
Istre B. [17]	1951	84	C	-	RO	-	-	-	-	-	Local metastasis	-	-	-	0	DOD
Istre B. [17]	1951	61	C	-	RO	-	-	-	-	-	Local metastasis	-	-	-	0	DOD
Istre B. [17]	1951	44	C	-	Bilateral	-	-	-	-	-	Local metastasis	-	-	-	0	DOD
Balasz M. et al. [18]	1960	60	C	-	RO	-	-	-	-	-	Local metastasis				0	DOD
Kelley RR. et al. [19]	1960	70	C	AP, nocturia	RO	106	-	BSO	NO	-	Abdomen	1	NO	NO	1	DOD
Numers Cv. et al. [20]	1960	70	C	Abdominal enlargement	RO	125	-	Supravaginal hysterectomy, remove mass, BSO and appendectomy	RT	-	-					-
De Gribble MG. et al. [21]	1962	46	C	AP	RO	120	-	no	NO	-	NO				0	DOD
Nieminen U. et al. [22]	1969	70	C	-	LO	50	-	Subtotal hysterectomy and BSO and appendectomy	RT	I	NO				80	NED
Azoury RS. et al. [23]	1971	66	-	-	-	-	-	-	-	-	-				1	DOD
Azoury RS. et al. [23]	1971	67	-	-	RO	-	-	-	-	-	Abdomen				24	DOD
Azoury RS. et al. [23]	1971	61	-	-	LO	160	-	-	-	-	Abdomen				16	DOD
Connor EJ. et al. [24]	1975	17	-	-	-	150	-	-	-	-	NO				12	NED
Walts AE. et al. [25]	1977	56	C	AP	LO	110	-	TAH, BSO	CT(NA) and RT	I	NO				7	NED
Reddy SA. et al. [26]	1985	48	C	AP	Bilateral	100	-	TAH, BSO and omentectomy	NO	III	NO				16	NED
Ansaldi E. et al. [27]	1986	60	C	AP, urinary frequency	RO	210	-	RSO, partial omentectomy	CT (adriamycin and Lomustine)	I	NO				10	NED
Cortes J. et al. [28]	1987	81	C	AP and constipation	LO	350	-	TAH, BSO	NO	I	NO				10	NED
Shakfen SM. et al. [29]	1987	NA	C	-	-	-	-	NA	NA	-	NO				12	DOD
Shakfen SM. et al. [29]	1987	24	C	-	-	-	-	TAH, BSO	RT	-	NO				48	DOD
Anderson B. et al. [30]	1987	59	C	-	-	-	-	BSO	CT (NA)	III	Local metastasis	18			18	DOD
Anderson B. et al. [30]	1987	45	C	-	-	-	-	USO	NO	I	Chest	36			54	DOD
Balaton A. et al. [31]	1987	35	C	AP	RO	120	-	TAH, BSO, omentectomy	CT (Adriblastine, vincristine, fluorouracil, cisplatin)	III	Peritoneum	7		RT	21	NED
Friedman HD. et al. [32]	1991	58	C	Lower AP and nausea	RO	260	-	TAH, BSO, PW	CT (dacarbazine, doxorubicin hydrochloride and cisplatin)	I	NO				9	NED
Nogales FF. et al. [33]	1991	32	C	AP	LO	-	-	no	NO	I	NO				36	NED
Nogales FF. et al. [33]	1991	48	C	Metrorrhagia and pain	LO	-	-	TAH, LSO, cytoreductive surgery, omentectomy	CT (ifosfamide and cisplatin)	III	Vagina and abdomen	9			24	DOD
Nogales FF. et al. [33]	1991	68	C		-	-	-	TAH, BSO, omentectomy	CT (cyclophosphamide, actinomycin D and vincristine)	III	NO				13	DOD
Monk BJ. et al. [34]	1992	12	C	Satiety, malaise, abdominal and back pain	LO	170	Normal	Exploratory laparoromy and BSO	Hormone therapy	I	NO					-
Dixit S. et al. [35]	1993	60	AA	Generalised weakness	RO	-	-	Excision of mass with TAH and BSO	RT on pelvis	IV	Lung	6	NO	NO	18	DOD
Rasmussen CC. et al. [6]	1996	32	C	Abdominal fullness	RO	130	35	TAH, BSO, partial ometectomy, PPLND, PW	CT (ifosfamide and mesna)	IIIC	Omentum, transverse colon	41	Optimally cytoreduced	NO	91	NED
Piura B. et al. [36]	1997	78	C	-	LO	150	-	NO	NO	IA	NO				21	Dead due to myocardial infarction
O’Sullivan SG. et al. [37]	1998	12	C	AP, malaise, fever, anorexia	RO	105	-	Laparotomy: RSO, omental biopsy, right iliac node biopsy, appendectomy	RT	-	YES	18				-
Nasu M. et al. [38]	1999	73	As	Difficulty in defecating	LO	170	Normal	TAH, BSO, omentectomy, dissection of regional lymph nodes	NO	I	Retroperitoneal	18			42	DOD
Rampaul RS. et al. [39]	1999	50	AA	AP	LO	150	-	TAH, BSO	NO	I	NO				24	NED
Inoue J. et al. [5]	2000	73	As	Constipation	-	170	Normal	TAH, BSO, PLND and omentectomy	NO	IIC	Liver, left kidney	16			42	DOD
Seracchioli R. et al. [40]	2002	20	C	NO	LO	80	Normal	LSO and multiple biposies	NO	-	NO				12	NED
Mayerhofer K. et al. [41]	2003	71	C	AP	LO	240	122	BSO and omentectomy	CT (cisplatin and ifosfamide)	IIIC	NO				14	Dead due to apoplexy
Kafah H. et al. [42]	2003	75	C	-	-	-	-	TAH, BSO, PPLND, omentectomy and appendectomy	CT	III	NO				24	NED
Lerwill MF. et al. [43]	2004	82	C	-	-	150				IA	NO				,	DOD
Lerwill MF. et al. [43]	2004	54	C	-	-	150				IA	Preauricular lymph node	24				-
Lerwill MF. et al. [43]	2004	66	C	-	-	48				IA	NO				28	NED
Lerwill MF. et al. [43]	2004	56	C	-	-	112				IA	NO				33	NED
Lerwill MF. et al. [43]	2004	55	C	-	-	220				IA	Lung and vertebrae				51	DOD
Lerwill MF. et al. [43]	2004	70	C	-	-	95				IA	Pelvic	47			63	DOD
Lerwill MF. et al. [43]	2004	25	C	-	-	80				IA	NO				48	NED
Lerwill MF. et al. [43]	2004	67	C	-	-	145				IA	Liver	49			73	DOD
Lerwill MF. et al. [43]	2004	42	C	-	-	80				IA	-				61	NED
Lerwill MF. et al. [43]	2004	69	C	-	-	300				IA	Pelvic	67				-
Lerwill MF. et al. [43]	2004	78	C	-	-	160				IA	NO				144	NED
Lerwill MF. et al. [43]	2004	53	C	-	-	140				IA	-					-
Lerwill MF. et al. [43]	2004	38	C	-	-	180				IA	Mediastinum, vertebrae	10			13	DOD
Lerwill MF. et al. [43]	2004	61	C	-	-	40				IA	NO				108	NED
Lerwill MF. et al. [43]	2004	65	C	-	-	130				IC	-					-
Lerwill MF. et al. [43]	2004	48	C	-	-	160				IIA	NO				12	DOD
Lerwill MF. et al. [43]	2004	70	C	-	-	-				IIB	Pelvic	4			6	DOD
Lerwill MF. et al. [43]	2004	70	C	-	-	90				IIB	NO				2	DOD
Lerwill MF. et al. [43]	2004	69	C	-	-	-				IIB	Lung	6			43	DOD
Lerwill MF. et al. [43]	2004	68	C	-	Bilateral	350		TAH, BSO and omentectomy		III	NO				13	DOD
Lerwill MF. et al. [43]	2004	47	C	-	Bilateral	130		TAH, BSO and omentectomy		IIIB	NO				3	DOD
Lerwill MF. et al. [43]	2004	61	C	-	-	180				IIIC	Pelvic	2			9	DOD
Lerwill MF. et al. [43]	2004	29	C	-	-	-				IIIC	NO				8	DOD
Lerwill MF. et al. [43]	2004	51	C	-	-	130				IIIC	-					-
Lerwill MF. et al. [43]	2004	49	C	-	-	175				IIIC	-					-
Lerwill MF. et al. [43]	2004	62	C	-	-	200				IIIC	NO				0	DOD
Nicotina PA. et al. [44]	2004	66	C	AP and abdominal enlargement	RO	140	-	TAH, BSO	CT (NA)	IIA	Liver, lung	18	NO	NO	24	DOD
Bouie SM. et al. [45]	2005	42	C	-	LO	180	-	Exploratory laparotomy and TAH with omentectomy	CT (NA)	I	NO				24	NED
Chang A. et al. [46]	2005	25	C	Fever, anorexia	RO	110	-	Excision of the mass	CT (doxorubicin, ifosfamide)	-	Large bowel	1	-	-	34	DOD
Kuscu E. et al. [47]	2005	62	C	Nocturia, Incontinence	RO	33	Normal	Laparoscopy: BSO. TAH, PPLNS	NO	I	NO				20	NED
Taskin S. et al. [15]	2007	68	C	NO	RO	120	Normal	TAH, BSO, PPLND, omentectomy and appendectomy	CT (ifosfamide and mesna)	IA	NO				118	NED
Taskin S. et al. [15]	2007	52	C	Abdominal distension	RO	-	-	TAH, BSO, PPLND, omentectomy and appendectomy	CT (ifosfamide and mesna)	IA	NO				19	NED
LI Y. et al. [48]	2008	71	As	AP and weight loss	LO	150	120.2	TAH, BSO, omentectomy, PLNS, peritoneal and pelvic wall biopsy	NO	III	NO				0	DOD
Khizar S. et al. [49]	2009	73	C	Painless	LO	140	Normal	BSO and omentectomy	CT, RT	-	-					-
Arslan OS. et al. [1]	2010	52	C	Inguinal pain	RO	80	139	TAH, BSO, infracolic omentectomy, appendectomy, bilateral PPLND	NO	IA	NO				6	NED
Dai Y. et al. [50]	2011	-	As	-	-	-	39.4	Optimal debulking	CT (cisplatin, VP-16/vincristine and bleomycin combination)	III	NO				4	DOD
Dai Y. et al. [50]	2011	-	As	-	-	-	39.4	Suboptimaldebulking	CT (cisplatin, VP-16/vincristine and bleomycin combination)	III	NO				4	DOD
Goodall EJ. et al. [4]	2011	60	C	Left loin pain, loss of appetite, sterile pyuria	LO	-	20	RH, BSO, omental biopsy, left PLNS	NO	IA	NO				22	NED
Zygouris D. et al. [51]	2011	58	C	AP	RO	250	63.4	Exploratory laparotomy:BSO, omentectomy, PLND	NO	IA	NO				21	NED
Pankaj S. et al. [52]	2013	27	AA	AP	RO	97	5.2	TAH, BSO, omentectomy, PPLND	CT (docetaxel and Gemcitabina)	-	NO				30	NED
Divya NS. et al. [9]	2014	26	AA	AP	RO	-	-	TAH, RSO	-	-	-					-
Rivas G. et al. [53]	2014	65	AA	AP and urinary retention	LO	150	Normal	TAH, BSO, PLND	CT (doxorubicin, ifosfamide and mesna)	IC	NO				24	NED
Sunita S. et al. [54]	2014	30	AA	AP	RO	150	69.9	Excision and omentectomy	CT (NA)						4	AWD
He M. et al. [14]	2015	46	As	AP	LO	50	27.73	Exploratory laparotomy: TAH, BSO	NO	IC	Pelvis, lung	11	Cytoreductive surgery	CT (docetaxel and gemcitabine (1 cycle) and gemcitabine only (2 cycles)	50	DOD
Kumar V. et al. [55]	2015	30	AA	AP	RO	147	69.9	Laparotomy: TAH, BSO, PW	CT (NA)	IA	NO				6	NED
Nazneen S. et al. [56]	2015	27	AA	AP and distension	RO	97	5.2	Laparotomy: TAH, BSO, omentectomy, PPLND	CT (docetaxel and gemcitabine)	-	NO				47	NED
Thyagaraju C. et al. [57]	2015	55	AA	AP and distension	RO	280	Normal	TAH, BSO, partial omentectomy	NO	I	NO				21	NED
Mamta G. et al. [58]	2015	27	AA	AP	RO	180	-	RSO, after TAH and LSO	NO	IA	NO				4	NED
Na Lee B. et al. [59]	2016	67	As	Gynecological checkup	RO	90	Normal	Adhesiolysis, TAH, BSO	NO	-	NO				3	NED
Pongsuvareeyakul T. et al. [13]	2017	65	As	AP	RO	242	202.5	Exploratory laparotomy: TAH, BSO, omentectomy	NO	IIIC	NO				1	DOD
Furutake Y. et al. [60]	2017	40	As	AP	LO	120	-	LSO	NO	-	LO, pelvic bone, liver	9	NO	CT (gemcitabine and docetaxel)	24	NED
Vishwanath. et al. [10]	2018	55	AA	AP, distension, loss of appetite	RO	150	150	TAH, BSO, PLND, omental biopsy	CT (docetaxel, gemcitabine)	IC	NO				2	NED
Tanaka A. et al. [61]	2018	64	As	Incontinence	LO	170	-	TAH, BSO, PLND, omentectomy	NO	IC2	Vaginal stump	7		CT (gemcitabine hydrochloride and docetaxel hydrate and then paclitaxel and carboplatin)	35	DOD
Sukgen G. et al. [62]	2018	59	C	AP and swelling	LO	350	Normal	TAH, BSO, BPPLND, omentectomy, PW, appendectomy	NO	IA	NO				6	NED
Shafiee MN. et al. [63]	2019	39	As	Heavy mestrual bleeding with clots and flooding, abdominal cramps and back pain	RO	200	40	TAH and BSO	CT (mesna, ifosfamide, doxorubicin and cisplatin)	IIIB	Lung	4	NO	CT (gemcitabine and paclitaxel)	24	NED
Fischetti A. et al. [64]	2019	61	C	Right colic hypochondrial pain	RO	90	-	Surgical resection	-	-	-					-
Yuksel D. et al. [65]	2020	34	C	-	RO	-	-	LSO, total omentectomy, PPLND, right ovarian resection and after RSO, TAH	CT (doxorubicin)	IA	NO				13	NED
Yuksel D. et al. [65]	2020	68	C	Pelvic pain, fever, sweats and rectal hemorrhage	RO	86	28	TAH, BSO, PPLND, omentectomy	CT (doxorubicin)	IIIC	NO				48	NED
Yuksel D. et al. [65]	2020	52	C	AP, distension	LO	-	-	PLND, omentectomy, tumor debulking, rectosigmoid resection	NO	IIIC	Early pelvic recurrence	1	NO	NO	2	DOD
Cojocaru E. et al. [2]	2021	-	C	-	-	-	-	TAH, BSO	Palliative CT	IVB	YES		NO	CT (ifosfamide nd doxorubicin)	25	DOD
Cojocaru E. et al. [2]	2021	-	C	-	-	-	-	TAH, BSO	NO	IA	NO		NO		15	NED
Cojocaru E. et al. [2]	2021	-	C	-	-	-	-	TAH, BSO	NO	IA	YES	14	NO	CT (doxorubicin)	51	DOD
Cojocaru E. et al. [2]	2021	-	C	-	-	-	-	TAH, BSO	NO	IA	YES	31	NO	CT (gemcitabine, docetaxel)	77	DOD
Cojocaru E. et al. [2]	2021	-	C	-	-	-	-	no	Palliative CT (doxorubicin)	IVB	YES	-	NO	CT (doxorubicin)	18	DOD
Cojocaru E. et al. [2]	2021	-	C	-	-	-	-	RSO	NO	IA	YES	16	NO	NO	98	AWD
Cojocaru E. et al. [2]	2021	-	C	-	-	-	-	TAH, BSO	NO	IA	YES	15	NO	CT (gemcitabine, docetaxel)	56	DOD
Cojocaru E. et al. [2]	2021	-	C	-	-	-	-	TAH, BSO	NO	IA	YES	3	NO	CT (carboplatin and gemcitabine)	22	DOD
Cojocaru E. et al. [2]	2021	-	C	-	-	-	-	LSO	NO	IA	NO		NO		8	NED
Cojocaru E. et al. [2]	2021	-	C	-	-	-	-	BSO	CT (gemcitabine, docetaxel)	IIB	YES	11	NO	CT (doxorubicin)	12	NED
Cojocaru E. et al. [2]	2021	-	C	-	-	-	-	NO	NO	IVB	NO		NO		2	DOD
Cojocaru E. et al. [2]	2021	-	C	-	-	-	-	BSO	NO	IA	YES	40	no	NO	47	AWD
Cojocaru E. et al. [2]	2021	-	C	-	-	-	-	LSO	NO	IA	NO				15	NED
Cojocaru E. et al. [2]	2021	-	C	-	-	-	-	TAH, BSO	NO	IIB	NO				13	NED
Cojocaru E. et al. [2]	2021	-	C	-	-	-	-	TAH, BSO	CT (gemcitabine, docetaxel)	IIIA2	YES	7	NO	CT (Epirubicin-carboplatin)	11	AWD
Pu T. et al. [16]	2022	29	As	Abdominal distension	RO	215	68.33	Exploratory laparotomy (fertility sparing)	NO	IA	LO	20	TAH, omentum resection, pelvic LA	CT (paclitaxel and carboplatin)	48	NED
Khadjetou V. et al. [66]	2022	16	AA	Pelvic pain	RO	150	Normal	TAH, BSO, PPLND	CT	IIIC	NO				23	NED
Bahadur A. et al. [67]	2022		As	Vaginal prolapse, lower abdominal pain	RO	150	55.6	TAH, BSO, omentectomy, small bowel resection, metastatectomy	CT (adriamycin and ifosfamide)	IIIC	-	-	-	-	-	-

C: Caucasian; AA: Afro-American; As: Asiatic; RO: Right ovary; LO: Left ovary; AP: Abdominal pain; TAH: Total abdominal hysterectomy; BSO: Bilateral salpingo oophorectomy; LSO: Left salpingo oophorectomy; RSO: Right salpingo oophorectomy; USO: Unilateral salpingo oophorectomy PW: Peritoneal washing; PLND: Pelvic lymphadenectomy; PPLND: Pelvic and paraaortic lymphadenectomy; PLNS: Pelvic lymph node sampling; PPLNS: Pelvic paraaortic lymph node sampling; RT: Radiotherapy; CT: Chemotherapy; DOD: Died of disease; NED: No evidence of disease; AWD: Alive with disease; NA: Not available; NO: the event did not occur.

**Table 2 cancers-15-02953-t002:** Summary table of the principal clinical characteristics of 113 primary ovarian leiomyosarcomas reported in the literature.

	Overall (*n* = 113)
AGE	
Mean (SD)	53 (12–84)
NA	18
**ETHNICITY**	
African	3 (2.8%)
Asian	22 (20.2%)
Caucasian	84 (77.1%)
NA	4
**SITE**	
Left Ovary	22 (34.9%)
Right Ovary	37 (58.7%)
Bilateral	4 (6.3%)
NA	50
**SIZE (mm)**	
Mean (SD)	151.2 (68.6)
NA	40
**Treatment**	
None	3 (3.8%)
Chemotherapy only	1 (1.2%)
Surgery	34 (42.5%)
Surgery with LND	10 (12.5%)
Surgery + CHT	21 (26.2%)
Surgery with LND + CHT	11 (13.8%)
NA	33
**Schedule of chemotherapy**	
Sarcoma-like	17 (70.8%)
Sarcoma-like +Platinum	5 (20.8%)
Dysgerminoma-like	2 (8.4%)
NA	9
**Adjuvant treatment**	
None	39 (50.0%)
Hormone therapy	1 (1.3%)
Radiotherapy	5 (6.4%)
Chemotherapy	31 (39.7%)
Chemotherapy + radiotherapy	2 (2.6%)
NA	35
**STAGE**	
I	52 (58.4%)
II	8 (9.0%)
III	25 (28.1%)
IV	4 (4.5%)
NA	24
**Recurrence**	
No	58 (56.9%)
Yes	44 (43.1%)
NA	11

**Table 3 cancers-15-02953-t003:** Multivariate linear model testing the effect of treatment, stage and mitotic count on the risk of death.

	Odds Ratio	C.I. 95%	*p* Value
**Surgery with LND**	1.3	0.8–1.6	0.284
**Surgery + CHT**	0.8	0.51–1.24	0.327
**Surgery with LND + CHT**	0.8	0.45–1.35	0.391
**Stage III-IV**	1.6	1.1–2.2	0.022
**Mitotic count > 10**	1.2	0.8–1.7	0.413

## Data Availability

The authors will provide the data when requested.

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
