# Peer review of "Primary Ovarian Leiomyosarcoma Is a Very Rare Entity: A Narrative Review of the Literature"

_cancers, 2023, doi:10.3390/cancers15112953_

Round 1

Reviewer 1 Report (Previous Reviewer 2)

The authors did a great work by revising their manuscript. The manuscript now is more scientifically sound and intellectually valid. All in all, the title of the review is clinically important and enriches the body of literature, and I expect it to be cited a lot in the future. I recommend acceptance in its current form.

Author Response

Dear Reviewer,

many thanks for your suggestions

Reviewer 2 Report (Previous Reviewer 1)

The authors have addressed the comments.

Appropriate.

Author Response

Dear Reviewer,

many thanks for your support

Reviewer 3 Report (New Reviewer)

This is a nicely written literature review of primary ovarian leiomyosarcoma (POLMS), which is a very rare gynecological malignancy. The authors include 113 cases from literatures or case reports since 1957, providing valuable information for researchers and doctors.

Author Response

Dear Reviewer,

many thanks for your favorable opinion

This manuscript is a resubmission of an earlier submission. The following is a list of the peer review reports and author responses from that submission.

Round 1

Reviewer 1 Report

In this review, the authors have analyzed published articles from 1951 to 2022 on primary ovarian leiomyosarcoma. The authors summarized the prevalence, mitotic counts, survival associated with surgical excision with lymphadenectomy and chemotherapy.

There are several issues with the review that need to be addressed:

1.     Simple summary statement is not provided. Instead, the original instruction is copied and pasted.

2.     The conclusion section in the abstract mentioned that the primary ovarian leiomyosarcomas are more common in younger women than epithelial ovarian cancer. This statement is confusing as the right comparison is not made. Either compare the statistics between primary ovarian leiomyosarcoma to epithelial ovarian cancer or between young versus older patients.

3.     Table 2 is rather misleading. It is unclear whether the race-associated prevalence is due to an actual ratio of the patient population or due to availability of samples.

4.     I understand the importance of studying primary ovarian leiomyosarcoma, but the manuscript is written in a way that is hard to understand. There are multiple data, but it is hard to extract what the takeaway messages are. It feels as though the manuscript was quickly put together. It would improve the quality of the manuscript if the storyline were organized.

5.     Minor suggestion: It would be helpful to have higher resolutions of Figure 1-4.  

Author Response

R1.

Comments and Suggestions for Authors

In this review, the authors have analyzed published articles from 1951 to 2022 on primary ovarian leiomyosarcoma. The authors summarized the prevalence, mitotic counts, survival associated with surgical excision with lymphadenectomy and chemotherapy.

There are several issues with the review that need to be addressed:

  1. Simple summary statement is not provided. Instead, the original instruction is copied and pasted.

Reply

We add simple summary as requested.

“Simple Summary: Primary ovarian leiomyosarcoma (POLMS) is a very rare malignancy characterized by unclear management and poor survival. We reviewed all 113 cases of POLMS reported in the literature till September 2022 to identify prognostic factors and the best treatment. Most patients received surgical resection, associated with lymphadenectomy in 12.5% of cases. Only 40% of patients received chemotherapy. POLMS is usually diagnosed at an early stage. Increasing stage and number of mitoses are associated with a worse prognosis. On the contrary, surgical resection with lymphadenectomy and chemotherapy is associated with increased survival. Ultimately, 43.4% of patients relapsed, and their mean disease-free survival was 12.5 months. There is a need for an international registry for POLMS that can help collect comprehensive and reliable data from around the world so that the best treatment can be definitively identified.

  1. The conclusion section in the abstract mentioned that the primary ovarian leiomyosarcomas are more common in younger women than epithelial ovarian cancer. This statement is confusing as the right comparison is not made. Either compare the statistics between primary ovarian leiomyosarcoma to epithelial ovarian cancer or between young versus older patients.

Reply

This statement was replaced with the following sentence “Primary ovarian leiomyosarcomas are more common in women in their 50s (mean age 53 years)”

Abstract: Background: Primary ovarian leiomyosarcoma is a very rare malignancy characterized by unclear management and poor survival. We reviewed all the cases of primary ovarian leiomyosarcoma to identify prognostic factors and the best treatment. Methods: We collected and analyzed the articles published in the English literature regarding primary ovarian leiomyosarcoma from January 1951 to September 2022, using PubMed research. Clinical and pathological characteristics, different treatments and outcomes were analyzed. Results: 113 cases of primary ovarian leiomyosarcoma were included. Most patients received surgical resection, associated with lymphadenectomy in 12.5% of cases. About 40% of patients received chemotherapy. Follow-up information was available for 100/113 (88.5%) patients. Stage and mitotic count were confirmed to affect survival, and lymphadenectomy and chemotherapy were associated with a better survival rate. 43.4% of patients relapsed, and their mean disease-free survival was 12.5 months. Conclusions: Primary ovarian leiomyosarcomas are more common in women in their 50s (mean age 53 years). Most of them are at an early stage at presentation. Advanced stage and mitotic count showed a detrimental effect on survival. Surgical excision associated with lymphadenectomy and chemotherapy are associated with increased survival. An international registry could help collect clear and reliable data to standardize the diagnosis and treatment.

  1. Table 2 is rather misleading. It is unclear whether the race-associated prevalence is due to an actual ratio of the patient population or due to availability of samples.

Reply

Table 2 is descriptive, therefore the distribution of the cases available in the literature is reported. In the study population most cases are described in the Caucasian population.

  1. I understand the importance of studying primary ovarian leiomyosarcoma, but the manuscript is written in a way that is hard to understand. There are multiple data, but it is hard to extract what the takeaway messages are. It feels as though the manuscript was quickly put together. It would improve the quality of the manuscript if the storyline were organized.

Reply

The manuscript was reviewed and the results highlighted.

  1. Minor suggestion: It would be helpful to have higher resolutions of Figure 1-4.  

Reply

The resolution of the figures has been improved as requested

Reviewer 2 Report

With pleasure, I read the paper titled: “Primary ovarian leiomyosarcoma a very rare entity: review of the literature”. Overall, the paper reads very well. The novelty lies in being among the first comprehensive review articles on the topic. The data are holistic, and the detailed analyses are major strengths of the report. The incorporation of radiographic and histopathological images adds beautification to the review article and provide important take-home educational messages to the readers. I congratulate the authors for a well-done project. I have the following comments/suggestions below. 

Title. Please revise to “Primary ovarian leiomyosarcoma is a very rare entity: a narrative review of the literature”.

Abstract. The first sentence in the conclusion subsection does not seem to be correct and please rectify: “Conclusions: Primary ovarian leiomyosarcoma are more common in younger women than epithelial ovarian cancers”.

Introduction. Highlight the gap in literature and the significance of your present work. Mention if a previous review has been completed in the past.

Methods. Your study design would have been much stronger if you would have performed it as a “systematic review” rather than “narrative review”. You searched only one database, and it is highly likely that you could have missed other cases in other important databases, such as Web of Science, Scopus, and Embase. You may want to highlight this point in your limitations section. You should provide better transparency of your data, by mentioning how many citations were retrieved from PubMed database, how many citations were omitted after title and abstracts, how many citations were subjected to full-text screening, and finally how many citations were finally found eligible for inclusion in your present study. This is better represented with a PRISMA diagram. If you can do this, it would be better. Please indicate the literature search was completed by how many authors, and how discrepancies were rectified (for example, by consensus between the authors or by discussion with the principal investigator). Similarly for the data collection, please indicate how many authors participated in the data collection, and how data were made sure to be correctly extracted. For statistical analysis, did you run normality test to check if you should use parametric or non-parametric test (such as the Shapiro-Wilk and Kolmogorov-Smirnov). For p values, were they one- or two-sided?

Results. Are there articles that reported case series, or all citations were just single case reports? For Table 1, it is recommend to add a column and add the reference if each study. For Table 2, it is important to mention the number of patients who had available data for each variable. For example, size of tumor was reported by how many cases? Also, for stage of tumor, you should include the number of cases with “unknown or not reported”. You should the same for variables in Table 2. For survival data mentioned in the texts, I do not see the HR values and the corresponding 95% CI values. They should be added. For Table 3, why did you use the odds ratio (OR) rather than the hazard ration (HR). Please justify, and not sure if the use of linear regression model is correct; it is supposed to be cox logistic regression; please double-check.

Discussion. The first paragraph should be a brief summary of the principal findings of your study. Please highlight the clinical implications and future directions. Please indicate the strengths and limitations of your research. Please compare and contrast your research with previous review articles, and highlight how your present research is difference from this one: Yuksel D, Cakir C, Kilic C, Karalok A, Kimyon G, Çöteli S, Boyraz G, Tekin ÖM, Turan T. Primary leiomyosarcoma of the ovary: a report of three cases and a systematic review of literature. J Gynecol Obstet Hum Reprod. 2021 Jun;50(6):101825. doi: 10.1016/j.jogoh.2020.101825. Epub 2020 Jun 1. PMID: 32497729. The last paragraph should be moved to a new section and titled “Conclusion”.

References. The references are not cited in order in the text. For example, in the introduction, the citations run from 1 to 8, and then suddenly jump to 16-18 and come back to 9 etc. Please correct.

Language. The manuscript will benefit from minor English polishing.

Author Response

R2.

Comments and Suggestions for Authors

With pleasure, I read the paper titled: “Primary ovarian leiomyosarcoma a very rare entity: review of the literature”. Overall, the paper reads very well. The novelty lies in being among the first comprehensive review articles on the topic. The data are holistic, and the detailed analyses are major strengths of the report. The incorporation of radiographic and histopathological images adds beautification to the review article and provide important take-home educational messages to the readers. I congratulate the authors for a well-done project. I have the following comments/suggestions below.

  1.  

 Please revise to “Primary ovarian leiomyosarcoma is a very rare entity: a narrative review of the literature”.

Reply

We revise the title as requested

  1.  

The first sentence in the conclusion subsection does not seem to be correct and please rectify: “Conclusions: Primary ovarian leiomyosarcoma are more common in younger women than epithelial ovarian cancers”.

Reply

This statement was replaced with the following sentence “Primary ovarian  leiomyosarcomas are more common in women in their 50s (mean age 53 years)”

  1.  

Highlight the gap in literature and the significance of your present work. Mention if a previous review has been completed in the past.

Reply

According to the reviewer's suggestion, we have reported in the introduction the differences of our review compared to those already present in the literature. We highlighted previous reviews that included only a small number of patients and did not address all possible risk factors. In our review instead, we have examined the English literature since 1951, trying to obtain the largest possible sample of patients and analyzing all the known prognostic factors, from clinical to histopathological ones, in order to be able to identify the best treatment and clarify the outcome of patients with POLMS.

  1.  

Your study design would have been much stronger if you would have performed it as a “systematic review” rather than “narrative review”. You searched only one database, and it is highly likely that you could have missed other cases in other important databases, such as Web of Science, Scopus, and Embase. You may want to highlight this point in your limitations section. You should provide better transparency of your data, by mentioning how many citations were retrieved from PubMed database, how many citations were omitted after title and abstracts, how many citations were subjected to full-text screening, and finally how many citations were finally found eligible for inclusion in your present study. This is better represented with a PRISMA diagram. If you can do this, it would be better. Please indicate the literature search was completed by how many authors, and how discrepancies were rectified (for example, by consensus between the authors or by discussion with the principal investigator). Similarly for the data collection, please indicate how many authors participated in the data collection, and how data were made sure to be correctly extracted.

Reply

We performed a literature review of the other databases and reported the selection criteria with a PRISMA diagram. We reported in the text who performed the literature review and how discrepancies were corrected.

“We collected and analyzed the published articles in the English literature regarding POLMS from January 1951 to September 2022, using Pubmed (https://pubmed.ncbi.nlm.nih.gov), SCOPUS (https://www.scopus.com/home.uri ), Web of Science (https://login.webofknowledge.com) research and the terminologies “primary ovarian leiomyosarcoma”, “primary leiomyosarcoma of the ovary”, “primary sarcoma of the ovary”, primary ovarian sarcoma”, “ovarian sarcoma”. Two authors performed the literature review and collected data. Discrepancies were corrected in discussions with the principal investigator, similarly correct data extraction was reviewed by the principal investigator. PRISMA (Preferred Reporting Items for Systematic Reviews and Meta-Analyses) flow chart with summary of search results is showed in figure 1. We identified 152 articles on Pubmed, 191 articles on Scopus, and 113 articles on Web of Science databases. After duplicates exclusion, 231 records underwent first-step screening of titles and abstracts. Of these, 74 full texts were considered for eligibility, and after reading them, 12 articles were excluded for being unfit according to the inclusion criteria or because they presented scant or aggregated data. Sixty-two studies were finally included in the review, for a total of 113 POLMS patients (Table 1).

  1. For statistical analysis, did you run normality test to check if you should use parametric or non-parametric test (such as the Shapiro-Wilk and Kolmogorov-Smirnov). For p values, were they one- or two-sided?

Reply

All continuous variable showed non-parametric distribution after Shapiro-test. P values were obtained by two-sides tests. We added these informations to “statistical analysis” paragraph

Results.

Are there articles that reported case series, or all citations were just single case reports? For Table 1, it is recommend to add a column and add the reference if each study. For Table 2, it is important to mention the number of patients who had available data for each variable. For example, size of tumor was reported by how many cases? Also, for stage of tumor, you should include the number of cases with “unknown or not reported”. You should the same for variables in Table 2. For survival data mentioned in the texts, I do not see the HR values and the corresponding 95% CI values. They should be added. For Table 3, why did you use the odds ratio (OR) rather than the hazard ration (HR). Please justify, and not sure if the use of linear regression model is correct; it is supposed to be cox logistic regression; please double-check.

Reply

The number of unknown (NA) data was reported for each variable in table 2.

HR and 95%CI were added in text in correspondence of figure 1A-B and 2A citation. For figure 3A the HR would be distorted due to the absence of event in the “Surgery with LND+CHT” group. For this reason we only reported the total pvalue and wrote “patients treated with surgery including lymph-adenectomy combined to chemotherapy showed a better prognosis and no event of death were registered during the follow up”. This is also the reason why in table 3 we applied a multivariate generalized linear model to test the effect of several variables on the risk of death, evaluating the number of events per group and not the time-dependent overall survival in order to obtain a more realistic statistical data.

Discussion.

The first paragraph should be a brief summary of the principal findings of your study. Please highlight the clinical implications and future directions. Please indicate the strengths and limitations of your research. Please compare and contrast your research with previous review articles, and highlight how your present research is difference from this one: Yuksel D, Cakir C, Kilic C, Karalok A, Kimyon G, Çöteli S, Boyraz G, Tekin ÖM, Turan T. Primary leiomyosarcoma of the ovary: a report of three cases and a systematic review of literature. J Gynecol Obstet Hum Reprod. 2021 Jun;50(6):101825. doi: 10.1016/j.jogoh.2020.101825. Epub 2020 Jun 1. PMID: 32497729.

 The last paragraph should be moved to a new section and titled “Conclusion”.

Reply

The last paragraph was moved to Conclusion section as suggested.

References.

The references are not cited in order in the text. For example, in the introduction, the citations run from 1 to 8, and then suddenly jump to 16-18 and come back to 9 etc. Please correct.

Reply

References in the text have been ordered as suggested.

Language. The manuscript will benefit from minor English polishing.

Reply

English were revised as requested.

Round 2

Reviewer 1 Report

In this review, the authors created a comprehensive literature review of primary ovarian leiomyosarcoma (POLMS) from January 1951 to September 2022, a total of 62 literature manuscripts (equivalent to 113 cases of POLMS). The authors analyzed the characteristics of POLMS reported in the literature and information about the effect of surgery, adjuvant therapy, risk factors, and markers on the overall survival probability of the patients. However, there are several modifications that need to make in this manuscript. 

  1. Describe in the abstract or in the introduction what exactly is primary ovarian leiomyosarcoma. 
  2. Figure 1: Explain why 157 records were excluded. 
  3. Table 1: What NO stands for? It is missing in the legend. 
  4. Lines 230 to 245: Explain why CA125, HE4, CEA, and mitotic counts are used as risk factors. 
  5. Lines 249 to 265: Explain what all these examinations are: SMA, Desmin, S100, etc. 
  6. Line 300: The authors mentioned that there is no significant association observed with the mitotic count. However, based on the figure, it looks very different. How is the p-value calculated? 
  7. Figure 4A, 4C, and 4D: The surgery+CHT value does not match the value in Figure 3A.
  8. Figure 4C: Figure 4C and 3B have 
  9. Figures 5B and 5D should have the same value for surgery with LND+CHT. The values for stages I-II and III-IV are flipped.
  10. Figure 5D: Also, the Surgery and LND values are flipped between the stages. It does not match the values in Figure 4.
  11. Line 450: Which specific platinum agents are used? Cisplatin? Carboplatin?

Minor edits:

  1. Please refrain from using the word, “useless” (line 55). I recommend using “ineffective”.
  2. Line 187: Include % after 77.1.
  3. Throughout the manuscript: Instead of saying “10/80 (12.5%) patients…” it reads better if it is written as “10/80 (12.5%) of patients…”
  4. Remove the word “strictly” in lines 299 and 311 as they are not strictly dependent on tumor grade. If it is, the value should be 100%. 
  5. Line 299: p = value 0.0066. Either change figure 3B (p=0.007) or the text (p=0.0066) to keep it consistent. 
  6. Figure 3: Include what CHT stands for in the figure legend. 
  7. Figure 3B and 3C: y-axis is missing.
  8. Figure 4: Include what CHT and LND stand for in the figure legend. 
  9. Figure 4B, 4C, 4D, 4E, and 4F: y-axis is missing.
  10. Table 3: Include what OR stands for in the figure legend. 
  11. Include A and B in Figure 6 and explain what the upper image and the lower image show exactly. 
  12. Figure 7: Include specifically what A, B, and C are showing in the figure legend. 
  13. Figures 8A and 9 are missing in the text. 
  14. Line 413: Include the abbreviation for SMA. 
  15. Line 486: Include the abbreviation for FOD.